# EvoSyn: Generalizable Evolutionary Data Synthesis for Verifiable Learning

## Abstract

Reliable verifiable data has become a key driver of capability gains in modern language models, enabling stable reinforcement learning with verifiable rewards and effective distillation that transfers competence across math, coding, and agentic tasks. Yet constructing generalizable synthetic verifiable data remains difficult due to hallucination-prone generation, and weak or trivial verification artifacts that fail to separate strong from weak solutions. Existing approaches often rely on task-specific heuristics or post-hoc filters that do not transfer across domains and lack a principled, universal evaluator of verifiability. In this work, we introduce an evolutionary, task-agnostic, strategy-guided, executably-checkable data synthesis framework that, from minimal seed supervision, jointly synthesizes problems, diverse candidate solutions, and verification artifacts, and iteratively discovers strategies via a consistency-based evaluator that enforces agreement between human-annotated and strategy-induced checks. This pipeline upgrades filtering into principled synthesis: it reliably assembles coherent, verifiable training instances and generalizes without domain-specific rules. Our experiments demonstrate the effectiveness of the proposed approach under both RLVR and model distillation training paradigms. The results show that training with our synthesized data yields significant improvements on both the LiveCodeBench and AgentBench-OS tasks, highlighting the robust generalization of our framework[1].

## 1 Introduction

Large language models (LLMs) have demonstrated remarkable potential across a wide range of domains, particularly in complex reasoning tasks such as mathematics, programming, and real-world agent applications. Recently, models like OpenAI-o1 and DeepSeek-R1 (Guo et al., 2025; OpenAI, 2024; Yang et al., 2025), after undergoing large-scale reinforcement learning, have shown significant improvements on reasoning benchmarks (Yue et al., 2025; Su et al., 2025). However, as model capabilities rapidly advance, their size continues to grow, and their demand for data is expanding at an astonishing pace. In particular, recent training paradigms increasingly rely on a special class of data—verifiable data.

Verifiable data provides reliable feedback signals during training, making it indispensable for many approaches. For example, RLVR-style training methods and model distillation heavily rely on such data (Schulman et al., 2017; Shao et al., 2024b; Zhao et al., 2025); DPO (Hosseini et al., 2024; Lai et al., 2024) leverages feedback to construct positive and negative samples; and various self-training methods such as STaR (Zelikman et al., 2022), V-STaR (Hosseini et al., 2024), and ReST (Singh et al., 2023) all depend on correctness signals to filter useful examples. However, the stringent reliability requirements of verifiable data make it extremely costly to annotate. Large-scale manual labeling is simply infeasible, highlighting the growing importance of verifiable data in modern LLM training pipelines.

Synthetic data offers a promising solution, but it remains imperfect (Liu et al., 2024; Long et al., 2024; Nadăs et al., 2025). Two persistent challenges limit its utility. First, *reliability*: hallucinations remain a fundamental weakness of LLMs. While models can generate large volumes of data, ensuring their reliability is nontrivial (Ding et al., 2024; He et al., 2025). How to make model-generated data

---

[1]We will release the code and data.

more reliable or how to effectively filter trustworthy subsets from large synthetic corpora remains a central challenge. Second, *generalizability*: Many existing solutions rely on task-specific, handcrafted heuristics to guarantee data usability. For example, some studies validate correctness through syntax checking (Wang et al., 2025). These approaches, however, often fail to generalize beyond the narrow task domains they were designed for.

In this work, we focus on these two questions: how to obtain reliable, verifiable data, and how to design a unified pipeline that generalizes across diverse tasks. We target the executably-checkable data class, which is the major part of verifiable data. We propose a general-purpose framework for synthesizing reliable data, called Evolutionary Data Synthesis (EvoSyn). Executably-checkable tasks are a broad class of problems defined as those for which verification can be performed via tests without requiring a complete solution. This class encompasses challenging real-world tasks, such as coding and software engineering problems. In our experiments, we select representative and high-difficulty tasks: the algorithmic LiveCodeBench (Jain et al., 2024) and the complex agent task AgentBench-OS (Liu et al., 2023). The core idea of EvoSyn is to formulate the difficulty as a data filtering strategy optimization task. Inspired by AlphaEvolve (Novikov et al., 2025), we employ evolutionary algorithms to iteratively search for the optimal filtering strategy tailored to the current task (Sharma, 2025; Romera-Paredes et al., 2024; Tanese, 1989). This strategy is then applied to synthetic data, yielding a reliable, verifiable dataset. Unlike prior approaches that require handcrafted, task-specific heuristics, EvoSyn automates this process: the model itself explores and evolves filtering strategies, reducing manual effort while producing superior solutions. Crucially, EvoSyn introduces a unified evaluation criterion for filtering strategies, which is task-agnostic. Instead of relying on domain-specific signals, EvoSyn measures consistency score with a small set of manually verified seed data, making it applicable to any verification task as long as minimal seed supervision is available.

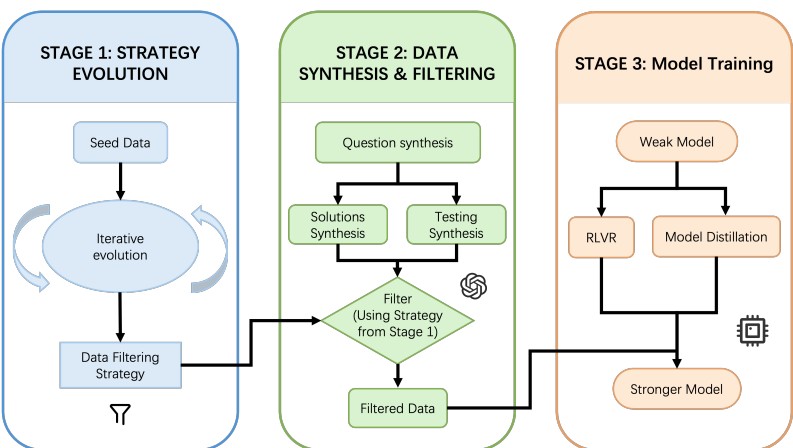

Figure 1: Overview of EvoSyn, a task-agnostic pipeline for synthesizing verifiable data. From a small human-verified seed data, an evolutionary process discovers a data-filtering strategy via a consistency-based evaluator; this strategy then guides synthesis by generating candidate solutions and tests for new problems, cross-executing them to rank and retain reliable instances while discarding trivial or inconsistent ones. The resulting verifiable dataset (problems, tests, and strong solutions) supports training in diverse tasks.

We demonstrate that EvoSyn is both effective and generalizable. Through its evolutionary process, EvoSyn continuously discovers novel and increasingly powerful strategies over iterations. We showcase representative examples and provide a detailed analysis of how strategy quality improves as the number of evolutionary rounds increases. Next, we validate EvoSyn on model training. On LiveCodeBench (Jain et al., 2024), we conduct RLVR training, and EvoSyn-generated data significantly improve the performance of LLaMA-3.1 (Grattafiori et al., 2024) and Qwen3 (Yang et al., 2025) models, outperforming raw synthetic baselines and providing more effective training dynamics. On the challenging AgentBench-OS benchmark, we choose the representative model distillation method, EvoSyn also yields substantial gains, enabling distilled models to surpass not only random baselines but also their teacher model (DeepSeek-R1 (Guo et al., 2025)).

Our main contributions are:

- We introduce Evolutionary Data Synthesis (EvoSyn), a general framework for synthesizing verifiable data. EvoSyn automatically evolves a superior data filterering strategies for the given task, enabling the construction of reliable synthetic datasets.

- We provide a detailed study of EvoSyn's evolutionary process, demonstrating its effectiveness, generalizability, and cost trade-offs.

- We validate EvoSyn on two important training paradigms, RLVR and model distillation, showing that EvoSyn-generated data yields substantial improvements over baselines.

## 2    RELATED WORK

**Verifiable learning**    Verifiable learning leverages executable or checkable feedback to supervise model training and spans both RL with verifiable rewards (RLVR) (Lambert et al., 2025) and supervised fine-tuning/distillation. In RLVR (Schulman et al., 2017; Shao et al., 2024b; Guo et al., 2025; OpenAI, 2024; Yang et al., 2025), correctness signals from program execution, unit tests, or other deterministic checkers stabilize training and markedly enhance reasoning ability. Beyond RLVR, teacher outputs can be filtered by execution in model distillation (Kim et al., 2025); and self-training pipelines such as RFT, STaR, and ReST (Singh et al., 2023; Zhang et al., 2024; Zelikman et al., 2022) rely on correctness signals to retain useful data. Verification feedback also constructs preference data for DPO (Hosseini et al., 2024; Lai et al., 2024; Rafailov et al., 2024) and improves reward models (Wang et al., 2023).

**Data synthesis**    Synthesizing verifiable data is critical yet challenging (Liu et al., 2024; Long et al., 2024; Nadăş et al., 2025). In practice, high-quality data for executably-checkable data often require broad-coverage unit tests (Chen et al., 2022a; Wang et al., 2025), program-analysis tooling (Liang et al., 2025), or carefully curated exemplars (Shao et al., 2024a). Such task-specific heuristics incur high manual costs and transfer poorly to complex real-world reasoning tasks (Fandina et al., 2025; Jimenez et al., 2023; Zhang et al., 2025a; Li et al., 2024). Hallucination further undermines reliability, making robust verification artifacts themselves a central bottleneck (Long et al., 2024).

## 3    METHODOLOGY

To address the inherent unreliability of synthetic data, we propose a new approach, *Evolutionary Data Synthesis (EvoSyn)*. EvoSyn targets executably-checkable tasks that satisfy two conditions: (1) correctness can be decided by executable "testing" artifacts (e.g., unit tests, checkers, environment assertions) that deterministically accept or reject candidate solutions; and (2) such testing artifacts can be authored without first producing a correct solution (e.g., via specifications, invariants, metamorphic relations, equivalence classes, boundary/edge cases). As illustrated in Figure 1, EvoSyn consists of three core stages: **(1) Deriving data filtering strategy**: deriving a reliable strategy using evolutionary algorithms. **(2) Data synthesis and filtering**: synthesizing data and filtering them with the derived strategy. **(3) Model training**: training models on the filtered synthetic data. The objective of EvoSyn is to establish a effective and automated mechanism that systematically enhances the reliability of synthetic data.

### 3.1    DERIVING DATA FILTERING STRATEGY

In the context of synthetic data, the central challenge is creating effective *testing* mechanisms that can reliably verify candidate solutions. A high-quality data instance typically consists of two components: a problem description and its corresponding testing set. Producing reliable testings is highly difficult because testings must not only reflect an understanding of the problem but also need to steadily distinguish correct from incorrect solutions. If a testing cannot differentiate solution quality, the instance becomes unreliable even if the problem itself appears well-formed. Therefore, improving the reliability of testings is the central focus of our method.

We require filtered data to satisfy two conditions: (1) the problem must be solvable, and (2) the testing must reliably distinguish correct from incorrect solutions. In practice, the second condition is more challenging. Reliable testings must consistently and correctly distinguish between correct and

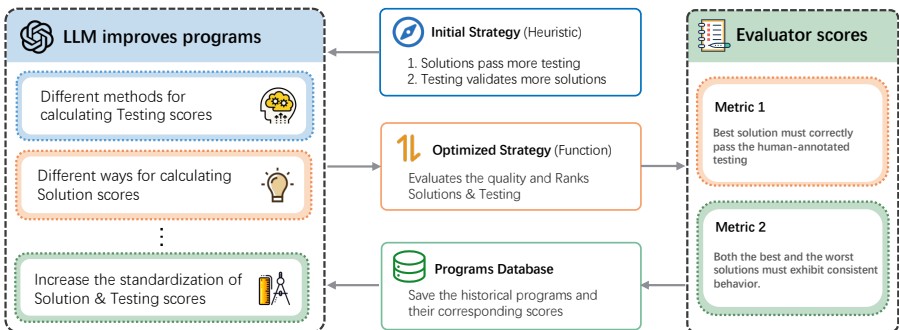

Figure 2: Given an initial strategy, the evolutionary algorithm iteratively optimizes it across multiple iterations. Each newly generated strategy is evaluated against our two criteria to determine its effectiveness. The model autonomously explores diverse optimization approaches, ensuring a balance between exploration and exploitation throughout the process.

wrong solutions, whereas proving that a problem is solvable only requires the existence of at least one solution that passes. Hence, reliability is the cornerstone of strategy design and optimization. We note that the core issue is how to select the best possible testing case from a theoretically unbounded set of generated candidates. To address this, we model the filtering strategy as a ranking function. The inputs to this function are a set of solutions and a set of testings, and the output is a ranked list of solutions (optional) and testings. *The top-ranked testing in this list is then selected as the final filtered testing.* By replacing the traditional filtering paradigm, which evaluates a single test and reduces it to a *boolean decision*, with a *ranking based formulation*, we are able to more effectively leverage the model's theoretically *unlimited generative capacity*. The ranking mechanism allows testings to be compared against one another, enabling the selection of the most optimal testing among a large pool of candidates. To derive such a strategy, we leverage seed data with human-annotated problems and testings. A relatively strong model is tasked with generating multiple candidate solutions for each problem, as well as additional testings based only on the problem description. These solutions and testings serve as inputs to the filtering strategy, which outputs a ranked list of solutions (optional) and testings.

The next question is how to obtain an optimally effective strategy function or program. Inspired by the success of Novikov et al. (2025), we adopt an evolutionary algorithm to iteratively improve the strategy. Evolutionary algorithms can balance exploration and exploitation. Following Novikov et al. (2025) and Sharma (2025), our implementation combines the MAP-Elites algorithm (Mouret & Clune, 2015) with island-based population models (Romera-Paredes et al., 2024; Tanese, 1989), enabling optimization over user-defined feature dimensions while maintaining population diversity. The overall workflow of the evolutionary algorithm is shown in Algorithm 1. After initializing the program to be optimized, the algorithm constructs a Database

---

**Algorithm 1:** LLM-driven Evolutionary Process

**Define** : Database $\mathcal{D}$, LLM $\mathcal{M}$, Evaluator $\mathcal{E}$,
PromptBuilder $\mathcal{S}$, Max Iterations $N$
**Output** : The best program $p_{\text{best}}$
**for** $n \leftarrow 1$ **to** $N$ **do**
    // Sample parent and inspirations
    $(p_{\text{parent}}, n) \leftarrow \mathcal{D}.\text{sample}()$;
    // Construct the prompt
    $\text{prompt} \leftarrow \mathcal{S}.\text{build}(p_{\text{parent}}, n)$;
    // Generate modification (diff)
    $\delta \leftarrow \mathcal{M}.\text{generate}(\text{prompt})$;
    // Apply diff to get child program
    $p_{\text{child}} \leftarrow \text{ApplyDiff}(p_{\text{parent}}, \delta)$;
    // Evaluate the child program
    $R \leftarrow \mathcal{E}.\text{execute}(p_{\text{child}})$;
    // Store result back to database
    $\mathcal{D}.\text{add}(p_{\text{child}}, R)$;
**return** $\mathcal{D}.\text{best}()$

---

that maintains multiple islands, each containing a population of candidate programs. In addition, the Database tracks a set of elite programs across islands. Importantly, the selection of elite programs considers not only evaluation scores but also factors such as code complexity, ensuring diversity within the population. During each evolutionary iteration, the algorithm simulates human-like evolution: it selects a parent program and provides it to the LLM for modification, together with several elite programs used as inspirations. Concretely, a PromptBuilder composes these historical programs into a structured prompt, which is then fed to the model to produce a new child program. The resulting child program is evaluated by an evaluator, which computes the consistency score described earlier.

This score serves as the performance indicator for the new program. The Database is then updated accordingly, and, after all evolutionary steps are complete, the algorithm returns the highest-scoring program as the final result. We design a initial strategy which *need not be optimal* according to two intuitive principles: (1) solutions that pass more testings are considered better; (2) testings that validate more solutions are considered better. Although this initialization is imperfect, for example, a testing that passes all solutions is likely uninformative, it suffices to bootstrap the evolutionary process, which will refine and correct such limitations. The evolutionary process also requires an *evaluator*, whose role is to assess strategy quality with respect to user-defined criteria. We define a good strategy as one that ranks testings in close agreement with human-annotated testings on seed data across diverse candidate solutions. Specifically, the method for evaluating the quality of a strategy includes two strict criteria:

- `Criterion-1`: the top-ranked solution produced by the strategy must correctly pass the human-annotated testing in the seed data.

- `Criterion-2`: for the ranked solutions, both the best and the worst solutions must exhibit consistent behavior on the annotated testing and on the best testing selected by the strategy. vs Figure 2 illustrates the actual workflow of our core method. Starting from an initial strategy, in each iteration the model explores various ways to obtain a better filtering strategy. From analyzing several relatively high-scoring strategies, we observe that the model explores multiple directions, such as refining the computation of solution quality and experimenting with different weighting schemes for testing. After each attempt, the evaluator assesses whether the new strategy satisfies our two predefined criteria on every instance in the seed data, and the proportion of satisfied cases is then used as the final score of the strategy, guiding the next round of evolution and refinement.

Remarkably, the evolutionary process yields multiple elegant and effective strategies. Figure 10 presents the best strategy evolved by model. The strategy scores each solution by the number of tests it passes, while testing scores are based on discriminative power (i.e., the gap between solutions' score that pass and fail), with both solution and testing scores normalized before computing discriminative power. Apart from this best one, model could explore various ways of computing testing scores. For example, *TF-IDF-like approach*: solutions that pass difficult testings receive higher scores, where difficulty is defined as testings passed by only few solutions; *Coverage-based approach*: solutions are rewarded simply for passing more testings, while testing quality is measured by its discriminative power (i.e., the score gap between solutions that pass and those that fail); *Inverse filtering approach*: contrary to the initial strategy, testings that fewer solutions can pass are considered better; *Exclusion-based approach*: the contribution of a testing is measured without its own influence, by weighting solutions that pass all other testings and *Hardness-Aware approach*: solutions are ranked by test strictness and pass count, penalizing all-or-none tests to select the strongest solution and most discriminative tests. The details of these strategies are illustrated in the Appendix A.3.

These evolved strategies demonstrate strong internal logic and significantly improve upon manually designed baselines, showing that evolutionary search can efficiently discover high-quality filtering strategies.

## 3.2 Data synthesis and filtering

With a robust filtering strategy in place, we proceed to data synthesis and filtering. Specifically, we first synthesize new problems to replace the human-annotated seed data. To ensure the generated problems are compatible with the filtering strategy, we provide seed instances as in-context examples to guide problem generation. After deduplication, the synthesized problems form a new set $D$. For each problem in $D$, we generate $M$ candidate solutions and $N$ candidate testings, which serve as inputs to the filtering strategy. The strategy ranks both solutions and testings. We then perform a final filtering step called *Zero-Variance Pruning*: we discard instances in which the testings yield no ranking variation. Such cases typically indicate either unreliable testings or trivial problems where all testings perform equally well. In both scenarios, discarding the instance is justified.

## 3.3 Model training

Following the above steps, we obtain a reliable synthetic dataset containing problem descriptions and their associated testings. As a byproduct, we also retain the strongest solutions generated by

the model. This dataset can be leveraged in various training paradigms, such as RLVR and model distillation, thereby boosting downstream model performance.

# 4 EXPERIMENTS

## 4.1 EXPERIMENTAL SETUP

This section presents a comprehensive set of experiments designed to validate the effectiveness of our proposed method. To address the specific challenge of verifiable problem synthesis, which is the core focus of our work, we conduct evaluations on two distinct and representative benchmarks: LiveCodeBench (Jain et al., 2024) and AgentBench-OS (Liu et al., 2023). LiveCodeBench is a highly challenging coding task benchmark, featuring a continuously updated collection of difficult programming problems. It has recently garnered significant attention within the large language model community due to its focus on real-time problem-solving capabilities. AgentBench-OS is a subset of the AgentBench benchmark, specifically designed to evaluate a model's performance in a realistic operating system environment. This benchmark assesses a model's ability to act as an intelligent agent and execute code to complete given tasks. The final performance is rigorously verified through a series of predefined tests.

To validate the effectiveness of our method, we conduct experiments on two representative training paradigms: reinforcement learning with verifiable rewards (RLVR) (Guo et al., 2025) and model distillation. Considering both cost and task complexity, we employ DeepSeek-V3[2] and DeepSeek-R1 as teacher models for the two tasks, respectively. The teacher model is responsible for the entire data synthesis pipeline, including the filtering strategy and data generation. During the evolutionary process for each task, we synthesize $M = 16$ solutions and their corresponding testing for every problem instance. The maximum number of evolutionary iterations is set to 20.

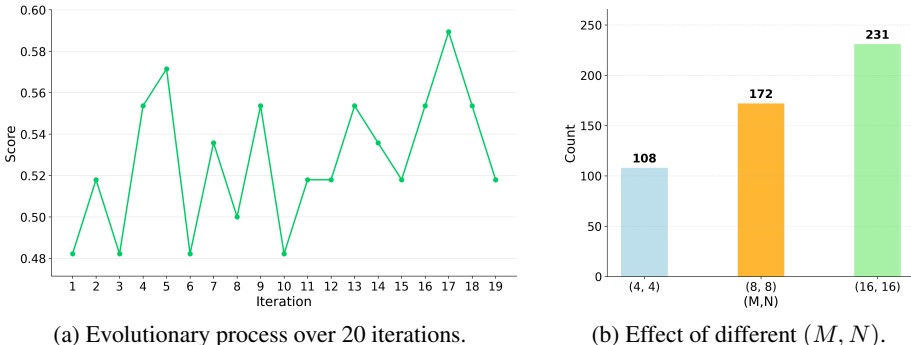

(a) Evolutionary process over 20 iterations.          (b) Effect of different $(M, N)$.

Figure 3: Evolutionary process and data-retention trade-off. (a) The evolutionary process consistently discovers stronger strategies, with the best strategy surpassing the initialization by over 10 percentage points within 20 iterations. Score denotes the ratio of seed data instances for which consistency verification is satisfied. (b) Increasing the number of $M$ and $N$ yields more usable, verifiable instances but incurs $O(MN)$ testing execution cost.

## 4.2 EVOLUTIONARY PROCESS

In this set of experiments, we use the LivecodeBench task as an example to illustrate the effectiveness of our core evolutionary method in the data synthesis process. As shown in Figure 3a, within the limit of 20 evolutionary iterations, and after excluding a few strategies that contained bugs, we frequently observe strategies outperforming the initial baseline. In particular, the best strategy exceeds the initial one by more than 10 percentage points, demonstrating both the model's ability to explore diverse strategies and the effectiveness of applying evolutionary algorithms to this problem. Moreover, the

---

[2]Due to the excessively long chains-of-thought (CoT) produced by DeepSeek-R1 on algorithmic problems, which lead to slow inference, we use DeepSeek-V3 as the teacher model for the LiveCodeBench task. We also observe poor performance in instruction following during question generation.

overall trend of the evolutionary process shows a steady upward trajectory. This suggests that, with more iterations, there is a strong potential to discover even better filtering strategies to guide data synthesis, highlighting the feasibility of our approach.

**Ablation study 1: Impact of $M$ and $N$**   However, better strategies also imply stricter filtering standards. To investigate this, we apply the best evolved strategy to data synthesis while varying the value of $M$ and $N$. The choice of $M$ and $N$ has a significant impact on synthesis cost, since our method requires generating $M$ solutions and $N$ testings, followed by $M * N$ executions. This quadratic growth in cost makes it crucial to understand the relationship between $(M, N)$ and the amount of usable data ultimately obtained. As shown in Figure 3b, when applying to the same set of 1,250 problems, using $M = N = 4$, $M = N = 8$, and $M = N = 16$ produces markedly different amounts of usable data. The reason is straightforward: with fewer samples, the likelihood of obtaining diverse solutions and testings decreases, making it harder to generate varied feedback and, consequently, to verify reliability.

**Ablation Study 2: Is Consistency Validation Sufficient with Only the Best and Worst Solutions?**   Recalling our two evaluation criteria for strategy assessment: `Criterion-1`, the best solution must be correct; `Criterion-2` the performance of the best and worst solutions must agree on both the human-annotated test set and the strategy-selected best test. A natural question arises: are these criteria sufficient? To investigate, we vary the number of solutions used for evaluating a same strategy (we use the initial strategy as an example) and choosing $M = 16$, considering the best and worst $K$ solutions with $K = 1, 2, 4$ and $K = 8$ (i.e., $M/2$). As shown in Table 1, increasing $K$ indeed strengthens the evaluation criteria, reflects in lower overall scores. However, two observations emerge. First, the stricter constraint does

Table 1: Consistency validation on $M = 16$ solutions. We vary $K$ and validate the same strategy using the top-$K$ and bottom-$K$ solution subsets. Adding `Criterion-1` at $K = 1$ yields the strictest check while requiring $8\times$ fewer executions (`#Exec`$= 4$ vs. $32$) than omitting it at $K = 8$. Increasing $K$ alone shows diminishing returns.

| $K$ | `Criterion-1` | Score | `#Exec` |
|---|---|---|---|
| 1 | No | 0.589 | 4 |
| 2 | No | 0.554 | 8 |
| 4 | No | 0.536 | 16 |
| 8 | No | 0.536 | 32 |
| *Ours* | | | |
| 1 | Yes | 0.482 | 4 |
| 8 | Yes | 0.482 | 32 |

not scale linearly with $K$: validating more solutions does not necessarily yield proportionally more accurate evaluations, largely due to randomness in solution sampling. Second, our setting combining the two criteria with $K = 1$, is in fact stricter than the $K = 8$ case, achieving both higher accuracy and significantly greater efficiency.

In our experiments, although the proposed method is in principle capable of generating unlimited data and producing highly reliable testings, from Figure 3b, we observe a log-linear relationship between the number of usable data instances and the number of testing executions. The underlying reason lies in the difficulty of controlling the diversity of model outputs. Low diversity inevitably requires larger values of $M$ and $N$, which substantially increases the cost. In future work, we aim to further investigate methods to enhance output diversity while reducing synthesis costs. In addition, practical bottlenecks such as slow model inference, time-intensive unit test verification, and costly environment setup further constrain the scalability of our data synthesis. We therefore adopt $N = 16$, yielding over 200 instances for LiveCodeBench and over 600 instances for AgentBench-OS. Despite this relatively small scale, training on these data still leads to substantial performance improvements.

## 4.3 PROBLEM-LEVEL ANALYSIS OF OUR DATASETS

Before verifying that our method can effectively filter out reliable and discriminative testings, we first examine the problem-level characteristics of the filtered datasets, including diversity/coverage and difficulty. For diversity and coverage, we compute the distribution of cosine similarities within both the filtered datasets and the original test sets for each task. We obtain embeddings using Qwen3-Reranker-0.6B (Zhang et al., 2025b), which allows us to assess how concentrated or diverse each dataset is. For difficulty, we evaluate Qwen3-8B, Qwen3-4B and Llama3.1-8B on both the original test sets and the filtered training sets. The resulting model scores serve as an indicator of the relative difficulty of each dataset. As shown in Figure 4, the filtered datasets exhibit a high degree of

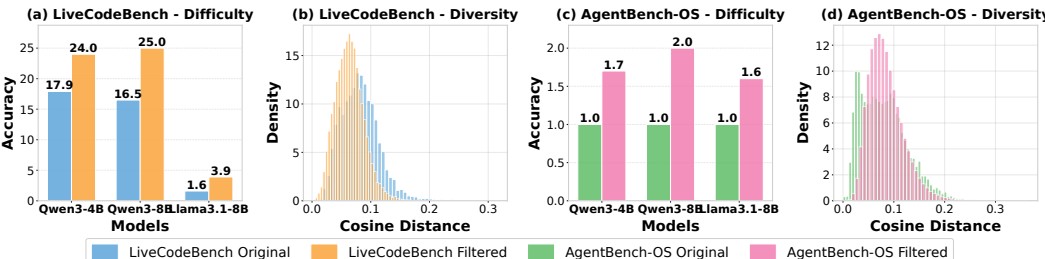

Figure 4: A comparison of the filtered datasets and the original datasets, at the problem level, in terms of diversity and difficulty for both tasks.

consistency with the original datasets in terms of problem diversity and coverage, displaying similarly broad distributions. Regarding problem difficulty, although the filtered datasets are somewhat easier than the original ones across different models, they remain highly challenging and leave substantial room for improvement. This ensures that, during training, the model does not encounter problems that are overly simple or insufficient to drive meaningful performance gains.

## 4.4 EVOSYN FOR RLVR

This set of experiments demonstrates that synthetic data generated with our method can effectively improve model performance in the RLVR task. We construct three data settings based on 51 seed instances:

- $D^{\text{EvoSyn}}$: Data filtered using our proposed data filtering strategy.

- $D^{\text{random}}$: Data with exactly the same problems as $D^{\text{EvoSyn}}$, but instead of using the filtering strategy, we randomly select one testing from the $N$ candidates as the final testing.

- $D^{\text{EvoSyn}^{\text{relaxed}}}$: Data obtained by relaxing the *Zero-Variance Pruning*, to investigate the necessity of our method's final filtering condition, which excludes instances that have not undergone ranking.

In particular, we analyze the number of unit tests per synthesized data in $D^{\text{EvoSyn}}$. As shown in Figure 9, the synthesized data contain an average of 11.5 unit tests, including various edge cases such as extremely long inputs. To mitigate the strong dependency on long-context capability imposed by such edge cases, we further adjust the testing generation process: instead of asking the model to directly produce unit tests, we require it to output code from which unit tests can be constructed. This not only preserves the diversity of unit tests but also ensures that the number of tests per problem remains sufficiently large.

Table 2: RLVR results on LiveCodeBench: Training on EvoSyn-filtered data ($D^{\text{EvoSyn}}$) consistently improves accuracy across models, outperforming random selection ($D^{\text{random}}$) and the relaxed variant ($D^{\text{EvoSyn}^{\text{relaxed}}}$). $\Delta$ denotes absolute gain over the baselines.

| Model | Data Setting | Dataset Size | Accuracy | $\Delta$ |
|---|---|---|---|---|
| *Baseline* | | | | |
| DeepSeek-V3 | - | - | 36.3 | - |
| Qwen3-4B | - | - | 17.0 | - |
| Llama-3.1-8B | - | - | 1.6 | - |
| Qwen3-8B | - | - | 16.5 | - |
| *RLVR Models* | | | | |
| Qwen3-4B | $D^{\text{EvoSyn}}$ | 231 | 22.0 | +5.0 |
| Qwen3-4B | $D^{\text{random}}$ | 231 | 19.9 | +2.9 |
| Llama-3.1-8B | $D^{\text{EvoSyn}}$ | 231 | 15.7 | +14.1 |
| Llama-3.1-8B | $D^{\text{random}}$ | 231 | 11.1 | +9.5 |
| Qwen3-8B | $D^{\text{EvoSyn}}$ | 231 | **24.8** | +8.3 |
| Qwen3-8B | $D^{\text{random}}$ | 231 | 21.1 | +4.6 |
| Qwen3-8B | $D^{\text{EvoSyn}^{\text{relaxed}}}$ | 256 | 24.4 | +7.9 |

**Results** We conduct reinforcement learning experiments on Qwen3-8B, Qwen3-4B, and Llama-3.1-8B using GRPO. As shown in Table 2, training on our synthesized dataset $D^{\text{EvoSyn}}$ consistently

yields significant performance gains across all models. Notably, on Llama-3.1-8B, we observe a substantial improvement of 14.1%. This demonstrates the effectiveness of our method in synthesizing reliable data.

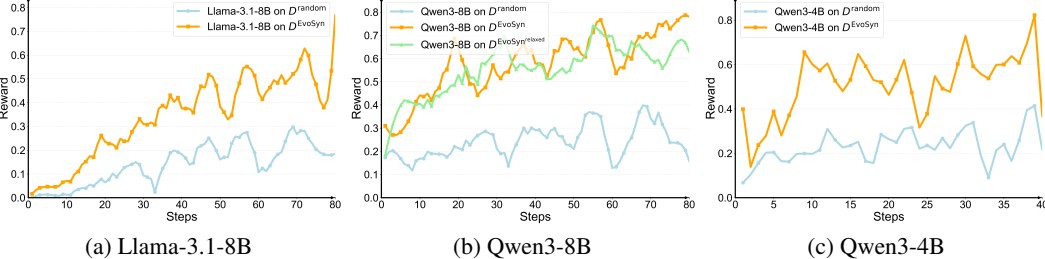

(a) Llama-3.1-8B          (b) Qwen3-8B          (c) Qwen3-4B

Figure 5: RLVR reward curves comparison across models. EvoSyn-filtered data ($D^{\mathrm{EvoSyn}}$) yields faster, steadier reward growth than random selection ($D^{\mathrm{random}}$).

**Ablation Study 3: What drives this advantage?** To further demonstrate the effectiveness of our method, we compare it against randomly synthesized data without filtering. As shown in Table 2, although training with randomly synthesized data on Qwen3-8B also yields some improvement, indicating that a portion of the data is indeed learnable, the performance still lags significantly behind that achieved with our filtered dataset. This result can be further analyzed through the reward dynamics during training. As illustrated in Figure 5, training on $D^{\mathrm{EvoSyn}}$ exhibits a steady and meaningful increase in reward, whereas training on $D^{\mathrm{random}}$ struggles to achieve consistent reward growth. This comparison highlights that the data constructed by our method is substantially more learnable for the model.

**Ablation Study 4: Is the *Zero-Variance Pruning* necessary?** In addition, we analyze the differences between $D^{\mathrm{EvoSyn}}$ and $D^{\mathrm{EvoSyn^{relaxed}}}$. By design, $D^{\mathrm{EvoSyn}}$ is a strict subset of $D^{\mathrm{EvoSyn^{relaxed}}}$. We manually examine the 25 additional instances present to $D^{\mathrm{EvoSyn^{relaxed}}}$ but not in $D^{\mathrm{EvoSyn}}$, and find that nearly all of them were overly simple problems. On average, their solution code lengths are only a dozen lines, and in some cases, the $M$ solutions sampled at temperature 1.0 are completely identical.

This observation validates the rationale behind the *Zero-Variance Pruning* in our method, which removes overly simple problems that provide little value for model learning. Such trivial problems are particularly problematic for the RLVR paradigm, as they prevent proper computation of the advantage. Selected example is provided in the Appendix A.4.

### 4.5 EvoSyn for Model Distillation

Model distillation has been widely adopted in the field due to its effectiveness and high efficiency, making it a powerful alternative to reinforcement learning, especially when the latter's training costs become prohibitive. Similar to RLVR, this method critically depends on a high-quality set of problems and reliable testings. This robust evaluation mechanism is essential for accurately filtering the correct responses

Table 3: Model distillation results on AgentBench-OS: EvoSyn-filtered data ($D^{\mathrm{EvoSyn}}$) yields large gains across students, outperforming random selection ($D^{\mathrm{random}}$). Remarkably, all students exceed the teacher (DeepSeek-R1, 30.1).

| Model | Data Setting | Accuracy | Δ |
|---|---|---|---|
| *Baseline* | | | |
| DeepSeek-R1 | - | 30.1 | - |
| Qwen3-4B | - | 1.0 | - |
| Llama-3.1-8B | - | 1.0 | - |
| Qwen3-8B | - | 1.0 | - |
| *Distilled Models* | | | |
| Qwen3-4B | $D^{\mathrm{EvoSyn}}$ | 40.0 | +39.0 |
| Qwen3-4B | $D^{\mathrm{random}}$ | 36.0 | +35.0 |
| Llama-3.1-8B | $D^{\mathrm{EvoSyn}}$ | 37.6 | +36.6 |
| Llama-3.1-8B | $D^{\mathrm{random}}$ | 22.0 | +21.0 |
| Qwen3-8B | $D^{\mathrm{EvoSyn}}$ | **44.9** | +43.9 |
| Qwen3-8B | $D^{\mathrm{random}}$ | 32.8 | +31.8 |

from a teacher model. In this experiment, we select the AgentBench-OS task, which is a highly realistic agent task requiring multi-turn, complex reasoning. These abilities are often a significant weakness for many models, particularly smaller ones. Due to the task's complex environment setup (the need for isolated Docker environments), the associated costs are prohibitively high, making RLVR-based training difficult. Therefore, we experimentally validate the effectiveness of our pro-

posed method within a model distillation pipeline. We use OpenHands (Wang et al., 2024) as the agent framework for our models. We filter the original AgentBench-OS dataset due to the presence of samples with stringent time requirements or permission issues. From the initial 144 data points, we retain 129, which are subsequently used as both the evaluation set and the seed data for our proposed method. This curation process ensures the reliability and reproducibility of our experimental results by focusing on a stable and accessible subset of the benchmark.

**Results**   Based on our method, we synthesize 673 data instances and obtain the corresponding outputs from DeepSeek-R1. Using this synthetic dataset, we train Qwen3-4B, Llama-3.1-8B, and Qwen3-8B. As shown in Table 3, all models exhibit substantial performance improvements after training. This not only highlights the weaker baseline performance of smaller models on complex, multi-turn, long-chain reasoning tasks but also clearly demonstrates the effectiveness of our synthetic data generation method. Furthermore, training on data synthesized by our method significantly outperforms training on randomly synthesized data, indicating that our approach is more effective at filtering usable data in complex, real-world agent tasks.

## 4.6 COMPARISON WITH BASELINES

We selected two representative baselines for comparison: LLM-as-a-Judge (Jiang et al., 2025) and CodeT (Chen et al., 2022b). Using an LLM as the source of evaluation signals has recently demonstrated strong performance across a wide range of tasks. In our setting, we provide the LLM with fine-grained testing evaluation criteria and require it to assign a score on a 100-point scale. CodeT is a representative method that identifies the best candidate through cross-execution, with its core grounded in a dual execution consistency algorithm. It relies on two key assumptions: 1. generated code solutions and test cases are independent and randomly sampled; and 2. the probability that two incorrect solutions coincidentally exhibit functional consistency is extremely low. CodeT evaluates solutions by identifying clusters of solutions that pass the same subset of testings, and uses the size of these clusters as the primary scoring metric.

Table 4: Comparison with baselines on LiveCodeBench and AgentBench-OS, including LLM-as-a-Judge and CodeT. Our method significantly outperforms the baselines on both tasks.

| Model | LiveCodeBench | | | AgentBench-OS | | | Avg. $\Delta$ |
|---|---|---|---|---|---|---|---|
| | Qwen3-8B | Qwen3-4B | Llama-3.1-8B | Qwen3-8B | Qwen3-4B | Llama-3.1-8B | |
| **Baseline** | 16.5 | 17.0 | 1.0 | 1.0 | 1.0 | 1.0 | |
| **Random** | 21.1 | 19.9 | 11.1 | 32.8 | 36.0 | 22.0 | 17.6 |
| **LLM-as-a-Judge** | 21.6 | 17.7 | 12.7 | 40.6 | 36.4 | 28.4 | 20.0 |
| **CodeT** | 22.4 | 16.8 | 15.1 | 43.3 | 39.1 | 27.2 | 21.1 |
| **EvoSyn** | **24.8** | **22.0** | **15.7** | **44.9** | **40.0** | **37.6** | **24.6** |

For each baseline, we applied its filtering procedure to the exact same synthetic dataset with our method, producing filtered datasets of identical size. We then trained models on these datasets using the same training configuration as in our method, and finally evaluated all models under identical testing conditions. We also include the results under the random setting here for comparison. As shown in Table 4, our method significantly outperforms the baselines on both tasks, indicating that our method is more effective at filtering usable data from large amount of synthetic data.

## 5 CONCLUSION

We introduce EvoSyn, a task-agnostic evolutionary data synthesis framework that focuses on synthesizing verifiable data for executably-checkable tasks by evolving robust filtering strategies from minimal seed supervision via a consistency-based evaluator. By turning ad hoc filtering into principled strategy optimization, EvoSyn assembles coherent, verifiable training instances that transfer across domains. On LiveCodeBench (RLVR) and AgentBench-OS (distillation), training on EvoSyn-filtered data yields substantial gains and superior learning dynamics across Llama-3.1-8B and Qwen3-4B/8B, with distilled students surpassing the teacher on complex multi-turn agentic tasks. Ablations confirm the value of strategy evolution and Zero-Variance Pruning, and characterize cost–quality trade-offs in $M * N$ execution. Limitations include verification/execution cost and output-diversity bottlenecks. Future work will scale population search, improve diversity-aware generation, and broaden verification tooling and domains.

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

# A  APPENDIX

## A.1  PROMPTS

This subsection provides the exact prompts used in our synthesis pipeline. We include the Live-CodeBench testing prompt and the AgentBench-OS prompts (There are two different types of responses, and we provide separate prompts for each type). They can be used to reproduce our data generation and to inspect task-specific constraints and formatting requirements.

```
I will give you a natural language description of a programming problem and you need to generate unit tests that
cover all edge cases of the problem.
Good unit tests should cover all inputs that are mentioned in the problem description, as well as any unique edge
cases that might not be obvious to the user.
The description of the problem might contains some simple unit test examples, but they are not enough to verify
the correctness of the solution. So you should refer to them to generate more comprehensive unit tests.
The format of the unit tests should be a strict json dict which can be loaded by json.loads() in python. And the
structure of the json dict should be a list of dicts, like following:

```json
[
{
"input": "input1",
"output": "output1"
},
{
"input": "input2",
"output": "output2"
}
...
]
```
The type of the input or output value should follow the type of the input or output value in the problem
description strictly.
The unit tests you generated should contain all the edge cases that are mentioned in the problem description, and
they also should follow the json format of the example unit tests.
The unit tests you generated should follow the constraints of the problem description strictly.
Sometimes the input of unit test may be very long, you can use a simple python code to generate the input like
Problem 3, the python code must can be process by the eval() function in python.
You should try to maximize the quality and coverage of the unit tests, here are some examples of good unit tests:
**Note**: Problem might contain a start code block, if so, you should provide the input parameters strictly in
accordance with the signature of the function, like Problem 1. But if there is no start code block, unit tests
must not contains any input parameters, your input and output should be a string, like Problem 2.

{examples}

Now please generate the unit tests for the problem.

## Problem
{problem}
```

Figure 6: Prompt for testing generation on LiveCodeBench.

```
Please give me the testing script for this task to get
the ground truth of the task.
The content you generate should be able to serve as the
content of an executable script. The execution result
of the testing script should be just the clean ground
truth of the task.
Please encapsulate your final testing script (script
content ONLY) within <testing> and </testing>.
For example: The testing script is <testing>
{qa_example} </testing>.

# Problem
{description}

# Environment Building Script
{init}
```

Figure 7: Prompt for testing generation on AgentBench-OS QA task.

## A.2  UNIT-TEST COUNT DISTRIBUTION

We analyze the number of unit tests attached to each synthesized problem to characterize the strength and granularity of our automated evaluation. Counts include both standard checks and long-input

```
Please give me the testing script for this task to judge
the correctness of the agent\'s execution solution.
The content you generate should be able to serve as the
content of an executable script. The execution result of
the testing script should be just the boolean value of
the correctness of the agent\'s execution solution.
Testing should get the result or effect of the agent\'s
execution solution first and then get the ground truth
of the task.
Finally, testing should compare the result with the
ground truth and output the boolean value of the
correctness of the agent\'s execution solution.
Please encapsulate your final testing script (script
content ONLY) within <testing> and </testing>.

# Problem
{description}

# Environment Building Script
{init}
```

Figure 8: Prompt for testing generation on AgentBench-OS Execution task.

edge-case tests. The distribution is broad (mean 11.5 per problem), indicating heterogeneous coverage and difficulty, which helps produce more stable and discriminative reward signals for ranking solutions and selecting tests.

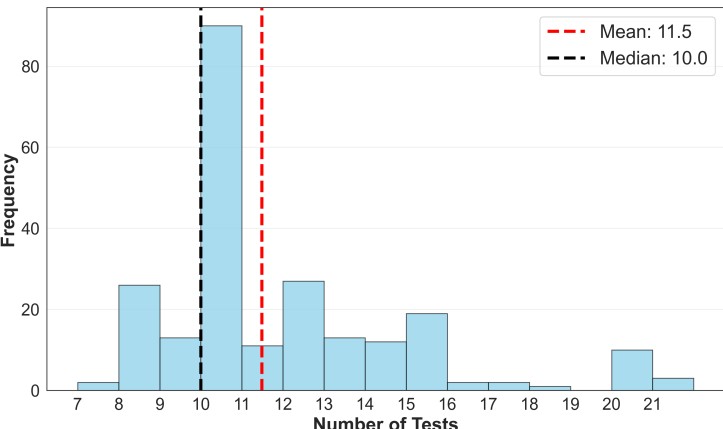

Figure 9: Unit-test count distribution per synthesized problem (mean 11.5); includes long-input edge cases.

### A.3 OPTIMIZED STRATEGIES

We outline several evolved scoring strategies that complement the best program in Figure 10: TF-IDF-like weighting, coverage-based scoring, inverse filtering, exclusion-based attribution, and hardness-aware ranking. Each aims to improve discriminative power and solution-ranking consistency on seed data.

```
1     # Compute solution_quality: fraction of testings passed by the solution.
2     total_testings = len(testing_id_list)
3     solution_quality = {}
4     for solution_id in solution_id_list:
5         count = len(solution_pass_testing[solution_id])
6         solution_quality[solution_id] = count / total_testings
7
8     # Compute testing_quality: difference between average solution_quality of passed vs
   failed solutions
9     testing_quality = {}
10    total_quality_all = sum(solution_quality.values())
11    total_solutions = len(solution_id_list)
12    for testing_id in testing_id_list:
13        passed_solutions = testing_accept_solutions[testing_id]
14        n_passed = len(passed_solutions)
15        total_quality_passed = 0.0
16        if n_passed > 0:
17            for sol_id in passed_solutions:
18                total_quality_passed += solution_quality[sol_id]
19            passed_avg = total_quality_passed / n_passed
20        else:
21            passed_avg = 0.0
22
23        n_failed = total_solutions - n_passed
24        if n_failed > 0:
25            total_quality_failed = total_quality_all - total_quality_passed
26            failed_avg = total_quality_failed / n_failed
27        else:
28            failed_avg = 0.0
29
30        testing_quality[testing_id] = passed_avg - failed_avg
31
32    # Sort solutions by solution_quality (descending) and testings by testing_quality
   (descending)
33    sorted_solutions = sorted(solution_id_list, key=lambda  : solution_quality[x],
   reverse=True)
34    sorted_testings = sorted(testing_id_list, key=lambda  : testing_quality[x],
   reverse=True)
```

Figure 10: The best strategy explored by model on LiveCodeBench.

```
1     # Compute n_T for each testing: number of solutions that passed the testing
2     n_T = {}
3     for testing_id in testing_id_list:
4         n_T[testing_id] = len(testing_accept_solutions[testing_id])
5
6     # Compute solution weights: using a TF-IDF like measure: weight of a solution is the
   sum of 1.0 / n_T for each testing it passed.
7     for solution_id in solution_id_list:
8         weight = 0.0
9         for testing_id in solution_pass_testing[solution_id]:
10            # If n_T[testing_id] is zero, skip to avoid division by zero (though it should
   be at least 1 since the solution passed it)
11            if n_T[testing_id] > 0:
12                weight += 1.0 / n_T[testing_id]
13        solution_weights[solution_id] = weight
14
15    # Compute testing_weights: 1/n_T for each testing, as a measure of how hard the
   testing is
16    testing_weights = {}
17    for testing_id in testing_id_list:
18        n = n_T[testing_id]
19        if n > 0:
20            testing_weights[testing_id] = 1.0 / n
21        else:
22            testing_weights[testing_id] = 0
```

Figure 11: TF-IDF-like approach. Solutions that pass difficult testings receive higher scores, where difficulty is defined as testings passed by only a few solutions.

### A.4 TRIVIAL PROBLEM

We manually examine the 25 additional instances present to $D^{\mathrm{EvoSyn}^{\mathrm{relaxed}}}$ but not in $D^{\mathrm{EvoSyn}}$, and find that nearly all of them were overly simple problems. Here we provide one of the examples, the quesiton is just a simple determination of whether some numbers are all even or not.

### A.5 AI USAGE STATEMENT

AI tools were used solely to assist with writing and polishing the main manuscript text. All core research content—including the ideas, problem formulation, methodology and algorithm design, data synthesis framework, experimental design and execution, implementation, evaluation, and analysis—was conceived, conducted, and validated exclusively by the authors. No AI systems were

```
1    # Compute solution weights: number of testings passed
2    for solution_id in solution_id_list:
3        solution_weights[solution_id] = len(solution_pass_testing[solution_id])
4
5    # Precompute total solution weight and total solutions for efficiency
6    total_solution_weight = sum(solution_weights.values())
7    total_solutions = len(solution_id_list)
8    # Compute testing weights: discrimination index (passed_avg_adjusted - failed_avg)
9    for testing_id in testing_id_list:
10        solutions_passed = testing_accept_solutions[testing_id]
11        n_passed = len(solutions_passed)
12        n_failed = total_solutions - n_passed
13        if n_passed == 0:
14            total_passed = 0
15            passed_avg = 0.0
16        else:
17            total_passed = sum(solution_weights[sol_id] for sol_id in solutions_passed)
18            passed_avg = (total_passed - n_passed) / n_passed   # subtract one for each
     passed solution (to remove the current testing) and average
19        if n_failed == 0:
20            failed_avg = 0.0
21        else:
22            total_failed = total_solution_weight - total_passed
23            failed_avg = total_failed / n_failed
24        testing_weights[testing_id] = passed_avg - failed_avg
```

Figure 12: Coverage-based approach. Solutions are rewarded simply for passing more testings, while testing qua[...]s that pass and those[...]

```
1    solution_weights = {solution_id: len(solution_pass_testing[solution_id]) for
     solution_id in solution_id_list}
2    total_solutions = len(solution_id_list)
3    testing_weights = {testing_id: total_solutions -
     len(testing_accept_solutions[testing_id]) for testing_id in testing_id_list}
```

Figure 13: Inverse filtering approach. Contrary to the initial strategy, testings that fewer solutions can pass are considered better.

involved in generating ideas, designing or running experiments, or producing any core research results.

```
1    # Compute solution weights: number of testings passed
2    for solution_id in solution_id_list:
3        solution_weights[solution_id] = len(solution_pass_testing[solution_id])
4
5    # Compute testing weights: average solution weight without the current testing for
     solutions that passed the testing.
6    for testing_id in testing_id_list:
7        solutions_passed = testing_accept_solutions[testing_id]
8        n = len(solutions_passed)
9        if n > 0:
10           total_weight = sum(solution_weights[sol_id] for sol_id in solutions_passed)
11           testing_weights[testing_id] = (total_weight - n) / n
12       else:
13           testing_weights[testing_id] = 0
```

Figure 14: Exclusion-based approach. The contribution of a testing is measured without its own influence, by weighting solutions that pass all other testings.

```
1    solution_weights = {solution_id: len(solution_pass_testing[solution_id]) for
     solution_id in solution_id_list}
2    total_solutions = len(solution_id_list)
3    testing_weights = {testing_id: total_solutions -
     len(testing_accept_solutions[testing_id]) for testing_id in testing_id_list}
4
5    # For each solution, compute the average testing weight (i.e., the average number of
     solutions failed by the testings that this solution passes)
6    solution_avg_testing_weight = {}
7    for solution_id in solution_id_list:
8        passed_testings = solution_pass_testing[solution_id]
9        if passed_testings:
10           avg_weight = sum(testing_weights[testing_id] for testing_id in
     passed_testings) / len(passed_testings)
11       else:
12           avg_weight = 0
13       solution_avg_testing_weight[solution_id] = avg_weight
14
15   # Sort solutions by: first the average testing weight (descending), then by the number
     of testings passed (descending)
16   best_solutions = sorted(solution_id_list, key=lambda x:
     (solution_avg_testing_weight[x], solution_weights[x]), reverse=True)
17
18   # Penalize testings that fail all solutions (broken testings) and also testings that
     pass all solutions (too lenient)
19   testing_penalties = {}
20   for testing_id in testing_id_list:
21       count = len(testing_accept_solutions[testing_id])
22       if count == 0:
23           # broken: reject all
24           testing_penalties[testing_id] = -100 * total_solutions
25       elif count == total_solutions:
26           # too lenient: accept all
27           testing_penalties[testing_id] = -50 * total_solutions
28       else:
29           testing_penalties[testing_id] = 0
```

Figure 15: Hardness-Aware approach. Solutions are ranked by test strictness and pass count, penalizing all-or-none tests to select the strongest solution and most discriminative tests.

```
1 You are given a positive integer N and a sequence of non-negative integers A = (A_1, A_2,
  ..., A_N) of length N.
2 Determine whether all elements in A are even numbers.
3
4 **Input**
5 The input is given from Standard Input in the following format:
6 N
7 A_1 A_2 ... A_N
8
9 **Output**
10 If all elements in A are even, print "Yes"; otherwise, print "No".
11 The judge is case-insensitive. For example, if the correct answer is "Yes", any of "yes",
   "YES", and "yEs" will be accepted.
12
13 **Constraints**
14 - 1 ≤ N ≤ 100
15 - 0 ≤ A_i ≤ 1000 (1 ≤ i ≤ N)
16 - All input values are integers.
17
18 **Sample Input 1**
19 3
20 2 4 6
21
22 **Sample Output 1**
23 Yes
24
25 All elements (2, 4, 6) are even numbers.
26
27 **Sample Input 2**
28 4
29 1 2 3 4
30
31 **Sample Output 2**
32 No
33
34 Not all elements are even (1 and 3 are odd).
35
36 **Sample Input 3**
37 5
38 0 0 0 0 0
39
40 **Sample Output 3**
41 Yes
42
43 Zero is considered an even number.
44
```

```python
1 n = int(input())
2 a = list(map(int, input().split()))
3 all_even = True
4 for num in a:
5     if num % 2 != 0:
6         all_even = False
7         break
8 print("Yes" if all_even else "No")
```

Figure 16: An example of a trivial problem present in $D^{\mathrm{EvoSyn}^{\mathrm{relaxed}}}$ but not in $D^{\mathrm{EvoSyn}}$, containing quesiton description and solution code.