# OpenReview forum: "EvoSyn: Generalizable Evolutionary Data Synthesis for Verifiable Learning"
_ICLR.cc/2026/Conference — ICLR 2026 Conference Withdrawn Submission_

### Official Review · Reviewer_gDs6 · 2025-11-01

**Soundness:** 3
**Presentation:** 3
**Contribution:** 3
**Rating:** 6
**Confidence:** 3

**Summary:**

This paper presents EvoSyn, an evolutionary data synthesis framework for constructing verifiable training datasets in tasks where correctness can be determined by executable tests. The method starts from a simple heuristic and then evolves data-filtering strategies, guided by a consistency-based evaluator that checks reliability on a small human-verified seed set. EvoSyn is evaluated on two executably-checkable domains — LiveCodeBench (for RL with verifiable rewards, RLVR) and AgentBench-OS (for model distillation). In both settings, EvoSyn-filtered data improve downstream model performance, leading to stronger reward learning and enabling smaller models to surpass their teachers.

**Strengths:**

- Treating reliable synthetic instance selection as a search over filtering strategies rather than fixed heuristics offers a clean, general abstraction applicable across verification-based learning setups.
- The two consistency-based criteria (ensuring solvability and discriminative tests) address the main causes of unreliable verifiable data, and the Zero-Variance Pruning step provides an efficient quality control mechanism.
- Applying the same pipeline to RLVR and distillation demonstrates strong generality.
- The paper clearly describes the limitations of handcrafted, task-specific test-synthesis heuristics and positions EvoSyn as a more automated alternative, though execution-cost limitations still constrain scale.

**Weaknesses:**

- Data scale is modest (231 RLVR; 673 distillation) from small seeds (51/129), in part due to the $O(MN)$ execution cost. The authors do not report variance across multiple evolutionary runs, so generality/reproducibility is hard to judge.

- Baselines are mostly intra-method (random/relaxed). Adding strong hand-designed verification baselines would clarify the benefit of evolution.

- The method selects for solvability and discriminativeness but does not report problem-level diversity/coverage/difficulty metrics. Quality is only inferred via downstream gains (and test-count statistics).

- Positioning vs prior evolutionary program/data-search work could be sharpened.

**Questions:**

- The authors report 231 (RLVR) and 673 (distillation) retained instances. Are these from a single evolutionary run/seed, or averaged over multiple runs?

- Given the $O(MN)$ execution cost, can the authors quantify the wall-clock/compute cost for the reported configuration and describe how much parallelism was used in practice?

---

> ### Author Response · Authors · 2025-11-20
> **Author's response to the weaknesses from Reviewer gDs6**
>
> ### **Weakness:**
>
> #### **W1: Data scale is modest (231 RLVR; 673 distillation) from small seeds (51/129), in part due to the $O(M*N)$ execution cost. The authors do not report variance across multiple evolutionary runs, so generality/reproducibility is hard to judge.**
>
> We sincerely appreciate the reviewer’s feedback. Indeed, due to the substantial computational overhead of our crossover operations, the version of the paper we submitted did not include a very large amount of generated data. The reviewer’s concern regarding potential fluctuations across multiple evolutionary runs is entirely valid.
>
> To address this, we conduct two additional evolutionary processes on the LiveCodeBench task. (We select this task because AgentBench-OS requires significantly more computational resources—specifically, the launch of large-scale Docker containers; see our response to Question 2 for details. Nonetheless, the conclusions remain consistent.) After obtaining two additional optimal strategies, we apply them to filter data using the __exact same__ synthetic dataset employed in the paper.
>
> We then compute the overlap between the datasets filtered by these two new strategies and the final dataset in the paper. The resulting overlap ratios are 97.8%, 92.6% and 95.2%, yielding **an average overlap of 95.2%**. This high degree of overlap demonstrates that the optimal strategies produced by our multi-round evolutionary procedure are highly consistent, thereby supporting the **generality and reproducibility** of our method.
>
> Conceptually, the optimal strategy for a given task tends to exhibit inherent similarity across runs. As the number of evolutionary rounds increases, the strategy function naturally converges toward a stable and consistent solution.
>
> #### **W2: Baselines are mostly intra-method (random/relaxed). Adding strong hand-designed verification baselines would clarify the benefit of evolution.**
>
> Thank you very much for raising this question. Indeed, comparing our method against strong baselines provides a clearer demonstration of its effectiveness. We surveyed a variety of data-filtering approaches for synthetic datasets and selected two representative methods from different methodological directions: LLM-as-a-Judge and CodeT, as our baselines. In **Section 4.6** of the updated paper, we present the comparative results in detail, which show that our method consistently outperforms both baselines.
>
> #### **W3: The method selects for solvability and discriminativeness but does not report problem-level diversity/coverage/difficulty metrics. Quality is only inferred via downstream gains (and test-count statistics).**
>
> Thank you for the reviewer’s question. Although our method primarily focuses on obtaining reliable and discriminative testings, it is indeed meaningful to also examine the problem-level characteristics. Accordingly, for both tasks we analyzed the diversity, coverage, and difficulty differences between the synthesized dataset and the original test set at the problem level. We provide a detailed presentation of our results in **Section 4.3** of the updated version of the paper.
>
> #### **W4: Positioning vs prior evolutionary program/data-search work could be sharpened.**
>
> We appreciate the reviewer’s suggestion. To the best of our knowledge, our work is the first to incorporate an evolutionary process into a data synthesis strategy. Nevertheless, we have added comparisons with representative baselines; please refer to our response in Section **W2** for details.

---

> ### Author Response · Authors · 2025-11-20
> **Author's response to the questions from Reviewer gDs6**
>
> ### **Question:**
>
> #### **Q1: The authors report 231 (RLVR) and 673 (distillation) retained instances. Are these from a single evolutionary run/seed, or averaged over multiple runs?**
>
> Thank you for raising this question. In fact, the evolutionary process is used **only** to search for the optimal data-filtering strategy. Once this optimal strategy is obtained, we apply it to the synthetic data to produce the final dataset. As demonstrated in our **W1** response, different evolutionary runs yield highly consistent results; therefore, a **single** evolutionary run is sufficient to obtain a reliable optimal strategy.
>
> It is worth noting that this high level of consistency is contingent on performing a sufficient number of evolutionary iterations. In our paper, for example, we set the number of iterations to 20. Generally, the more iterations performed, the more consistent the resulting optimal strategies will be; conversely, too few iterations may lead to divergence.
>
> #### **Q2: Given the $O(M*N)$ execution cost, can the authors quantify the wall-clock/compute cost for the reported configuration and describe how much parallelism was used in practice?**
>
> Thank you for the reviewer’s concern regarding computational cost. In our experiments we set $ M = N = 16$, which implies that each problem requires $M \times N = 256$ verification executions. During this project, we utilized a machine equipped with 150 CPU cores. For LiveCodeBench we are able to run 256 concurrent processes, yielding an average throughput of approximately one problem per minute—an overall processing speed we consider acceptable.
>
> By contrast, AgentBench-OS is a multi-round, complex agent task that requires strict environmental isolation: each execution must run in its own Docker container to ensure consistency. Consequently, for each problem we must maintain 256 independent containers. Due to the high memory footprint per container, our practical concurrency for AgentBench-OS was 64, which substantially reduces throughput relative to LiveCodeBench; on average, each problem takes roughly ten minutes to complete. The final dataset of over 600 examples reported in the paper was produced after approximately two weeks of continuous execution and filtering, and this extended runtime is the primary reason the dataset size is limited.

---

### Official Review · Reviewer_9iJM · 2025-11-02

**Soundness:** 3
**Presentation:** 2
**Contribution:** 2
**Rating:** 4
**Confidence:** 4

**Summary:**

The paper proposes EvoSyn, a task-agnostic evolutionary data synthesis framework designed to generate verifiable synthetic data for LLM training. Verifiable data (i.e., data with executable correctness checks) is crucial for reinforcement learning with verifiable rewards (RLVR) and distillation, yet remains expensive to curate manually and hard to generalize across domains.

EvoSyn tackles this by evolving data filtering strategies that identify reliable problems, solutions, and verification artifacts (tests) through an evolutionary optimization process guided by consistency with a small human-verified seed dataset. The method iteratively refines filtering strategies based on two strict criteria ensuring alignment between human-annotated and model-inferred correctness.

Once a high-quality filtering strategy is obtained, EvoSyn synthesizes new tasks, candidate solutions, and tests, filters them using the evolved strategy, and trains models using this curated data. Experiments on LiveCodeBench (coding tasks, RLVR setting) and AgentBench-OS (agentic reasoning tasks, model distillation setting) demonstrate gains. EvoSyn-filtered data substantially improves model performance, enabling smaller distilled student models to outperform teacher models.

**Strengths:**

- The approach of synthesizing verifiable data is domain-agnostic, contrasting prior heuristic or task-specific filtering methods. Its evolutionary optimization of filtering strategies is broadly applicable.

- The framework is evaluated on two different benchmarks (LiveCodeBench and AgentBench-OS) under both RLVR and distillation paradigms, showing performance gains.

- The paper decomposes the pipeline (strategy evolution, synthesis, filtering, training), provides detailed ablations (e.g., effect of M, N, criteria sufficiency, pruning), and articulates trade-offs between data reliability, diversity, and computational cost.

- The paper includes prompts, strategy variants, and explicit evaluation criteria in the appendix, thus focusing on reproducibility.

**Weaknesses:**

- The paper is dense but not well-written and well-structured. For instance, throughout the introduction, the authors repeatedly emphasize developing a general framework for synthesizing verifiable data, yet the exact task formulation and problem statement remain vague. The objective is presented at a very high level without clearly defining the input-output structure of the task. Only by examining the experimental setup and the prompts in the appendix does it become apparent that the core task is test-case generation from NL problem descriptions. These descriptions, similar to those in competitive programming problems, may contain a few example test cases while the exhaustive test suite remains hidden. The framework also asks the LLM to generate several candidate solutions. Subsequently, EvoSyn performs cross-execution of the generated solutions and tests, for example, using TF-IDF-like, coverage-based, inverse filtering, or exclusion-based scoring approaches.
However, since both the candidate solutions and test cases are generated by LLMs, they may both be unreliable or semantically inconsistent with the original NL description. How do the authors ensure that the generated test cases are meaningful and semantically faithful to the input description, rather than reflecting coincidental or spurious correlations?

- The paper introduces a TF-IDF-like scoring mechanism where solutions that pass "difficult" tests receive higher scores, with difficulty defined as tests that are passed by only a few solutions. However, the underlying task—NL description to test-case generation, makes this assumption problematic. If a test case is passed by only a few solutions, it does not necessarily indicate that it is difficult; rather, it could simply be faulty or semantically misaligned with the problem description. In the absence of ground-truth verification or semantic alignment checks, there is no clear justification for treating such cases as valuable or discriminative. This undermines the reliability of the evolved scoring strategies and calls into question whether the "difficulty" metric genuinely correlates with test-case quality.

- "For example, RLVR-style training methods...": please define any abbreviation before using it for the first time

- There are typos in the paper e.g., "synthsizing" in line 146

- For the prompt provided in Figure 6, what are the "Problem 1", "Problem 2", "Problem 3" being referred to? These are not defined anywhere in the prompt provided in Figure 6.

- Instead of providing code-snippets in the paper and appendix, I would suggest providing algorithms that are typically more reader-friendly.

- The metrics in the experimental section are not clearly defined. How is "accuracy" in Table 3 computed?

- The evaluation covers only two benchmarks and model families. Given that the paper mentions that it focuses on developing a "general framework" for synthesizing verifiable data, broader tests on other verifiable domains (math reasoning, data-to-text, scientific QA) would better demonstrate true generality.

- EvoSyn relies on a small set of human-verified seed data to guide consistency-based evaluation. The paper does not deeply explore how biases or poor coverage in this seed data affect the evolved strategies.

**Questions:**

See weaknesses

---

### Official Review · Reviewer_vbBU · 2025-11-11

**Soundness:** 2
**Presentation:** 2
**Contribution:** 2
**Rating:** 2
**Confidence:** 3

**Summary:**

The paper describes a method to generate verifiable synthetic data along with an automatic evolution based filtering technique to select useful datapoints from it. The evolution based filtering technique works by initializing a simple strategy and evolving it while trying to 'fit' to a small human-annotated dataset of synthetic datapoints. The overall approach works in 3 stages: (a) evolve a strategy iteratively (b) ask a strong model to generate problems, solutions and tests/oracles and filter them using the evolved strategy (c) train the model on this data. The paper evaluates this approach on LiveCodeBench using RLVR and on AgentBench-OS using distillation, and shows that the filtering helps train better models compared to randomly sampling synthetic data.

**Strengths:**

The paper proposes an approach to filter synthetic data without relying on task-specific heuristics, and evaluates the approach on two common approaches of post-training (RLVR and model distillation), and on two benchmarks providing some evidence of the generality of this approach.

**Weaknesses:**

**Baseline**: The paper does not compare the approach with real baselines. The baselines used in the paper are simple/artificial. What would be great is if the paper can compare against other filtering approaches (heuristics or other automatic approaches) so as to compare the efficacy of this filtering approach over other filtering/synthetic-data-generation approaches. Being better than random baseline is not very meaningful as it is expected that randomly generated data without any kind of filtering will be very noisy, what would be a meaningful claim is if you can show that you get similar benefits as SOTA task-specific-heuristics without having to actually manually define them yourself.

Relatedly, the "related works" section is also very sparse and the paper could benefit from broadening it significantly and actually comparing it with the approach in this paper.

**Method clarity**: The other major concern I have is that the methodology is not entirely clearly described. E.g., how exactly is the strategy evolution happening? Perhaps you could describe the algorithm in more details/using pseudocode. An example of something that's not clear from the writing: how can a strategy that outputs a *ranked* list of (solutions, tests) for every problem be used for *filtering* the data (filtering is a boolean function) -- are you filtering based on a cutoff? How are you picking problems that go in the training set based on this ranked list? Just formally writing down the process would clarify the method significantly, and would greatly improve the paper.

**Questions:**

Apart from the above major concerns I raised, I have a question about the criteria -- doesn't satisfying criterion 2 imply that criterion 1 is automatically satisfied?

---

> ### Author Response · Authors · 2025-11-20
> **Author response to Reviewer vbBU**
>
> ### **Weakness:**
>
> #### **W1: Baseline: The paper does not compare the approach with real baselines. The baselines used in the paper are simple/artificial. What would be great is if the paper can compare against other filtering approaches (heuristics or other automatic approaches) so as to compare the efficacy of this filtering approach over other filtering/synthetic-data-generation approaches. Being better than random baseline is not very meaningful as it is expected that randomly generated data without any kind of filtering will be very noisy, what would be a meaningful claim is if you can show that you get similar benefits as SOTA task-specific-heuristics without having to actually manually define them yourself.**
>
> Thank you very much for raising this point. Comparing our method against representative baseline approaches provides a much more effective demonstration of its efficacy. We surveyed various synthetic data filtering methods and selected two distinct, representative approaches as our baselines: **LLM-as-a-Judge** and **CodeT**. We have detailed the comparative results in **Section 4.6 of the updated version of our paper**. The results consistently show that our method maintains a stable advantage over both of these baselines.
>
> #### **W2: Method clarity: The other major concern I have is that the methodology is not entirely clearly described. E.g., how exactly is the strategy evolution happening? Perhaps you could describe the algorithm in more details/using pseudocode. An example of something that's not clear from the writing: how can a strategy that outputs a ranked list of (solutions, tests) for every problem be used for filtering the data (filtering is a boolean function) -- are you filtering based on a cutoff? How are you picking problems that go in the training set based on this ranked list? Just formally writing down the process would clarify the method significantly, and would greatly improve the paper.**
>
> We sincerely thank the reviewer for raising this problem. In response, we have thoroughly revised the description of the evolutionary algorithm in the paper. We have also made fine-grained updates to the algorithmic workflow illustrated in **Figure 2**. Specifically, we have added detailed pseudocode (Algorithm 1), updated the workflow diagram, and provided a clearer explanation of why our filtering strategy is formulated as a **ranking function** and how this function is applied during data filtering at **Section 3.1**.
>
> ### **Question:**
>
> #### **Q1: Apart from the above major concerns I raised, I have a question about the criteria -- doesn't satisfying criterion 2 imply that criterion 1 is automatically satisfied?**
>
> Thank you for the reviewer’s question. These two criteria are indeed central to our method: together they define the metric used to measure a strategy’s consistency score. The rationale behind their design is as follows. We require that testing selected by a strategy produce behavior that is *consistent* with the ground-truth testing when evaluated across different solutions. In the paper we evaluate this by comparing the outcomes of the best and worst solutions, which, relative to the ground-truth test suite, yields four possible combinations (each solution may be correct or incorrect).
>
> Fundamentally, the two criteria do **not** have a containment or hierarchical relationship. It is entirely possible that the best solution is incorrect (thereby violating Criterion 1), yet the testing produced by a strategy still behaves consistently with ground-truth testing. The reverse scenario can also occur. Only strategies that satisfy **both** criteria simultaneously are awarded a positive consistency score.
>
> We hope this explanation clarifies the reviewer’s concern.

---

### Author Response · Authors · 2025-11-20
**General response to all reviewers**

We thank all reviewers for their thoughtful questions and comments. We have responded to every weakness and concern raised, and have made corresponding revisions to the paper (marked in red). These revisions include: an expanded explanation of the algorithmic workflow and an updated flowchart in Section 3.1; an added problem-level analysis of the synthesized dataset in Section 4.3; and new comparisons with representative baselines in Section 4.6.

We would be very grateful for the reviewers’ further feedback, and we hope that our responses successfully address their concerns. Once again, we sincerely appreciate the reviewers’ time and careful consideration.

---

### Note · Authors · 2026-01-06

I have read and agree with the venue's withdrawal policy on behalf of myself and my co-authors.